# Sago Haemolytic Disease: A Case Due to Climate-Induced Food Insecurity in Western Province, Papua New Guinea

**DOI:** 10.3390/tropicalmed6040190

**Published:** 2021-10-23

**Authors:** Mikaela Seymour

**Affiliations:** Rural Medical Education Australia, Middle Fly District, Balimo 360, Western Province, Papua New Guinea; mikaela_seymour@hotmail.com

**Keywords:** sago palm, haemolytic disease, clinical treatment, low-resource medicine, tropical medicine, Papua New Guinea, Western Province

## Abstract

Sago haemolytic disease (SHD) is a rare but significant condition presenting in sago starch-eating populations in Western Province, Papua New Guinea. Although rare, case fatality rates are high, and no known antidote is available. The exact cause of the disease is unknown, but it is believed to be secondary to mycotoxins produced by fungi in old sago. In this case report, a 50-year-old female was treated in a low-resource setting in Middle Fly, with fluid resuscitation and transfusion, making a full recovery without complications. The mainstay of treatment for SHD is intravenous fluid resuscitation and strict fluid balance, which can be achieved in even the most remote Western Province aid post. Increased food insecurity, secondary to climate change, may see the incidence of this condition increase. Therefore, all health workers in Western Province should be comfortable with fluid resuscitation and fluid balance practices.

## 1. Introduction

Sago starch, refined from the sago palm tree (Metroxyon Sagu), is commonly eaten throughout Southeast Asia and Melanesia. Papua New Guinea is the second largest grower of the crop [1]. The sago palm is heavily relied upon as a primary source of carbohydrate energy, with up to 300 kg of starch harvested from each tree [1]. Few diseases can harm the sago palm, and it grows in locations and conditions in which other crops cannot grow [1]. The high yield of the sago palm, its ease of growth and perennial harvesting have led to calls to increase sago palm crops throughout the tropics [1]. In Western Province, Papua New Guinea, the consumption of sago starch is ubiquitous [2,3].

A description of sago haemolytic disease (SHD) was first published in 1974 [2,3]. The exact cause of the condition is unknown; however, mycotoxins are heavily suspected [2]. It is believed that aerobic conditions of storage permit fungi to grow, which unsuspecting families consume [2]. The fungal hyphae are broken down during digestion, which causes a free fatty acid excess in communities low in albumin (secondary to protein insufficiency malnutrition) [2]. Excess of free fatty acids destroys the red cell membrane, causing intravascular haemolysis [2].

Environmental conditions, such as flooding and drought, lead to food insecurity and are becoming increasingly common [4]. Food insecurity causes populations to eat older sago, which is more likely to contain dangerous fungi. As climate conditions continue to change, food insecurity is expected to increase, potentially leading to an increase in cases of SHD.

Patients often present 12–24 h after consuming the spoiled sago, reporting headache, fever, dizziness, weakness, vomiting, jaundice and haematuria [2,5]. Western Province has the highest rate of SHD worldwide, despite the fact that 300 million people internationally consume sago starch regularly [1,2]. The Middle Fly District recorded the greatest number of outbreaks, with three registered between 1998 and 2005 [2]. The incidence of SHD is estimated at 5.8/100,000 in Western Province, with a case fatality rate of 35% over a 10-year period [2]. The case fatality rate may be artificially high as it is likely that very remote cases are not adequately reported. Although genetic disposition in this region has been considered a potential risk factor, investigation of the effect of Melanesian ovalocystosis by Rai suggested it is unlikely to be significant [6]. There is no standardised national treatment guideline for SHD; however, intravenous fluids and transfusion are the most common interventions [2].

Publications thus far have focused on the aetiology of SHD and potential preventative behaviours to stop its occurrence. Although these studies are crucial, they do not assist health workers who are faced with SHD at the clinic. This case report aims to provide primary health care workers with a context and framework to initiate good clinical care to minimise morbidity and mortality in patients suffering from SHD.

## 2. Case Presentation

A 50-year-old female Lake Murray resident presented at the Boboa Health Centre with spontaneous gross haematuria. Her family reported a 24 h history of increasing abdominal pain and agitation with confusion. The patient denied diarrhoea, vomiting, dysuria or coryzal symptoms preceding presentation. This history was confirmed by the sister who lived with the patient. She was previously well, with no known medical history. The patient’s sister reported the family had eaten ‘old sago’, harvested three months earlier, in the two days preceding the presentation. The sago was stored in a bucket in the home.

Initially, the 50-year-old mother was the only member of the family who presented with haematuria. Within 24 h, the patient’s 12-year-old daughter and 5-year-old grandson also presented with painless haematuria but nil abdominal pain or confusion. In total, three of the five family members who consumed the Sago developed gross haematuria. Only the mother required admission. Further history revealed the mother had prioritised the sago starch, which looked and felt fresher, for her children, consuming the older-looking sago herself. She also consumed a larger quantity of the starch.

The nursing officer at Boboa reported nine similar cases presenting in the last three months alone; however, they were all members of one family. Three of these cases required referral to the district hospital for transfusion, and one death was reported. In addition, the health worker reported old sago starch had been consumed due to ongoing, unprecedented flooding in Lake Murray and the inability to use the water for new sago preparation due to faecal contamination by flooded pit latrines.

## 3. Examination, Investigations, Diagnosis, Treatment and Outcomes

On presentation, the examination findings demonstrated a soft but tender abdomen with nil signs of peritonism. The pain was localised to the suprapubic region and right and left lower quadrants. There were no organomegaly and no signs of jaundice. The patient was agitated and often crying in pain, unable to follow commands and not orientated to the surroundings. The patient could not walk due to generalised ‘body weakness’, with neurological examination revealing 4/5 strength in all four limbs and sensation intact. The pupils were equal and reactive to light. The Glasgow Coma Scale was 14 (due to confusion). The chest was clear, and heart sounds were dual with nil murmur detected.

Her vital signs were as follows: heart rate, 74 beats per minute; blood pressure, 130/80 mmHg; temperature, 37.3 degrees Celsius; and oxygen saturation, 94%. After 12 h of observation, these vitals were as follows: heart rate, 95 beats per minute; blood pressure, 100/70 mmHg; temperature, 38.5 degrees Celsius; and oxygen saturation, 92%.

Due to being in remote Western Province, Papua New Guinea, investigations were limited. However, on the day of presentation, the malaria rapid diagnostic test was negative. The malaria rapid diagnostic test was repeated 12 h later (when the patient developed fever) and was still negative. The Haemocue portable haemoglobin machine recorded an initial haemoglobin level of 8.5 g/dL, which decreased to 7.3 g/dL and 6.8 g/dL at 12 and 18 h measurements, respectively. On gross examination, the urine was rosé in colour (see Figure 1). Urine dipstick demonstrated haemoglobin ca. 50, leucocytes ca. 75 and protein 100 and was negative for nitrites. Blood glucose was 4.6 mmol/L measured with the AccuCheck portable blood glucose machine.

The initial most likely diagnosis was urinary tract infection (with associated cystitis, pyelonephritis and/or renal calculi). Malaria, despite the negative rapid diagnostic test, and SHD were a differential diagnosis. SHD is incredibly rare, but the nine previous cases in this community gave it increased weight as a potential cause of the presentation.

A urinary catheter was inserted, and intravenous fluids were administered with strict fluid balance. The patient received 4.5 L of fluid over 18 h, whilst draining 5 L in the catheter bag. She was monitored for clinical signs of fluid overload. The patient was commenced on 1 g paracetamol per rectum, and per oral cotrimoxazole 960 mg as per the Standard Treatment Manual for Papua New Guinea treatment for cystitis. Twelve hours later, the patient showed no improvement and was commenced on intravenous ceftriaxone 1 g. Despite these antibiotic interventions, the pain and agitation continued, and gross haematuria did not abate. The fever was reduced with administration of paracetamol.

At the twelve-hour mark, two further members from the same family, who had consumed the same sago starch, presented with painless gross haematuria, making the diagnosis of SHD the most likely cause.

The 50-year-old female was transferred from the remote Boboa outpost to Kiunga District Hospital where fluid balance continued. Unfortunately, their laboratory was out of reagents and was unable to provide haemoglobin on arrival. The physician administered one unit of fresh blood based on clinical signs. The haematuria ceased after 24 h at the hospital (72 h since onset). The patient was discharged 5 days later after full recovery.

## 4. Discussion

The lack of awareness campaigns, or funded programs, to tackle dangerous sago storage mechanisms means cases of SHD will continue to present to health workers in Western Province. Urgent referral to a hospital is the preference for management of SHD patients; however, logistics in remote areas mean medivacs are often delayed or not possible. Health workers need to be confident to provide the best management possible in a low-resource setting.

Health workers within Western Province should be mindful that SHD remains rare, and for patients presenting with haematuria, other more likely differential diagnoses such as urinary tract infection/cystitis should be considered first [7]. Malaria is also much more common in Western Province, and severe malarial haemorrhagic fever can lead to haematuria [8].

Despite the case fatality rate, health workers should be reassured that the condition is self-limiting and likely based on the quantity of fungal hyphae ingested. There is no known antidote to SHD, and management should focus on supportive care similar to the principles used for other haemolytic crises [2]. This involves restoration of total haemoglobin to above 5 g/dL (if transfusion is safely available), monitoring of tissue oxygenation, strict fluid balance, adequate hydration via intravenous fluid support, body temperature control with antipyretics and observation for metabolic acidosis [9].

The two potential complications of SHD which likely cause death are severe haemolytic anaemia and kidney failure. Acute haemolysis can cause acute renal failure, secondary to deposition of haem pigmentation, leading to tubular necrosis [10]. This can be avoided by dialysis [10]. Dialysis is only available in Western Province at Tabubil Hospital. In low-resource settings, in the absence of dialysis, intravenous fluids should be administered to maintain the intravascular volume, promoting perfusion of the renal system, and assisting in the excretion of haemolysis waste products, decreasing their deposition in the renal system [11].

Intravenous resuscitation and fluid balance can be performed in remote centres and do not require specialist equipment, where the aim should be to continue the resuscitation until the urine is clear and the patient’s symptoms have resolved. Frusemide is also available rurally and could be used to treat low urine output, increasing blood flow to the renal system, but should be discontinued if there is no diuretic response within three hours [12]. This should only be administered with the support of a medical officer.

## 5. Conclusions

In conclusion, SHD remains a rare but serious condition for the people of Western Province, Papua New Guinea. Health care workers in Western Province need to be mindful of the condition as a potential differential diagnosis and provide appropriate supportive care to reduce morbidity and mortality in this low-resource setting.

## Figures and Tables

**Figure 1 tropicalmed-06-00190-f001:**
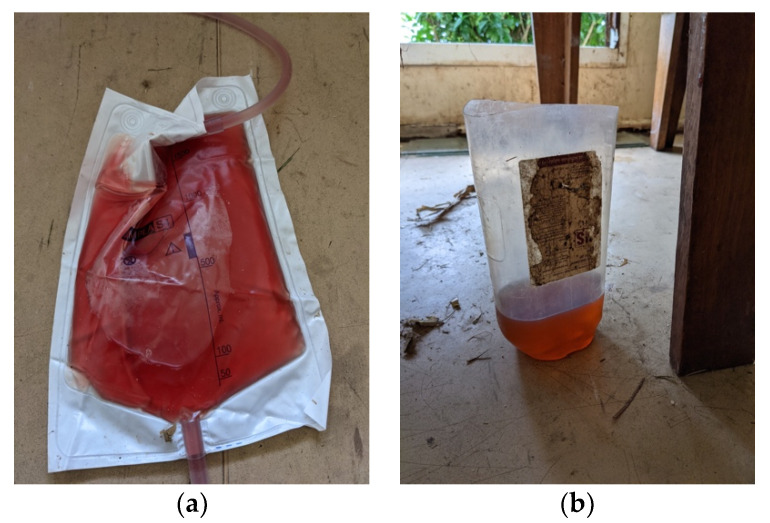
(**a**) Haematuria as seen once the indwelling catheter was placed in the patient; (**b**) the initial sample of urine provided by the patient on presentation.

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
