# Peer review of "Sago Haemolytic Disease: A Case Due to Climate-Induced Food Insecurity in Western Province, Papua New Guinea"

_tropicalmed, 2021, doi:10.3390/tropicalmed6040190_

Round 1

Reviewer 1 Report

As this is a case study the level of detail provided in the introduction and particularly the conclusion, while brief, is adequate. Check manuscript for typos- e.g. line 61 'minimis' should be 'minimise', line 54 'standardisd' should be 'standardised'

Line 126- Sentence starting '12-hours' should be 'Twelve hours'

Institutional Review Board Statement may require more information - often ethical approval is not required for case studies- similar statement could be included. 

Author Response

Thank your Reviewer 1. 

I have made the changes suggested. I have no concerns with the proposed suggestions. 

Thank you for your time reviewing this paper. 

Reviewer 2 Report

Overall, this is an interesting and well written case report that aims to provide basic guidance to clinicians that may encounter SHD. The evidence that this case is, in fact, SHD is well presented and compelling. I have no major concerns, but following are several minor points that should be addressed prior to publication.

  1. Please include time period for the incidence of SHD in Western Province (Line 50-51).
  2. Hematuria due to organophosphate poisoning is extremely rare (case reportable), so unless the author has a good quality reference to support the stated association please remove (Line 119).
  3. The reported case fatality (Line 155) certainly overstates the actual case fatality given under ascertainment of mild cases, which is likely to be substantial, would decrease the CFR. As such, please highlight that the 35% CFR is based on reported cases, and the actual CFR is likely to be lower.
  4. Lastly, and most importantly, it’s not clear where the recommendation to transfuse for Hgb<8.0 g/dl originated. The reference provided (#9, for NTDT) states to not transfuse unless Hgb<5 g/dl. Transfusing blood in very low resource settings is a relatively high-risk intervention, given reagents and quality controls for screening and cross matching are often lacking. Therefore, unless patients are markedly symptomatic due to low oxygen carrying capacity or actively hemorrhaging, the transfusion threshold for otherwise healthy individuals should be high.

Author Response

Thank you reviewer 2 for your careful consideration of this paper. 

Point 1. Happy to accept this suggestion, have modified to reflect the reference, the incidence of 5.8/100,000 was over 10 years

Point 2. Happy to remove this reference. 

Point 3. Happy to insert this context, I accept the CFR is likely lower with cases going unreported and therefore skewing CFR. 

Point 4. Thank you for this suggestion. Happy to change to 5 g/dl to reflect the reference. Agree transfusion should not be taken lightly in low resource settings. Will adjust to reflect the lower Hb and the inclusion of clinical correlation with signs of anaemia. 

Thank you for your review.